# A Fast Algorithm to Simulate Nonlinear Resistive Networks

**Benjamin Scellier**
Rain AI
`benjamin@rain.ai`

## Abstract

Analog electrical networks are explored as energy-efficient platforms for machine learning. In particular, resistor networks have recently gained attention for their ability to learn using local rules such as equilibrium propagation. However, simulating these networks has been challenging due to reliance on slow circuit simulators like SPICE. Assuming ideal circuit elements, we introduce a fast simulation approach for nonlinear resistive networks, framing the problem of computing its steady state as a quadratic programming (QP) problem with linear inequality constraints. Our algorithm significantly outperforms prior approaches, enabling the training of networks 327 times larger at speeds 160 times faster.

## 1 Introduction

As energy costs for machine learning continue to rise, neuromorphic computing platforms are gaining attention as alternatives to traditional neural networks and GPUs (Momeni et al., 2024). These platforms use analog physics and compute-in-memory architectures to achieve substantial energy savings. Among them, nonlinear resistive networks have sparked particular interest (Kendall et al., 2020; Dillavou et al., 2022, 2024; Wycoff et al., 2022; Anisetti et al., 2024; Stern et al., 2022, 2024; Kiraz et al., 2022; Watfa et al., 2023; Oh et al., 2023). These networks use variable resistors (e.g., memristors) as trainable weights, combined with diodes for nonlinearity and voltage/current sources as inputs. Nonlinear resistive networks can be trained by gradient descent via equilibrium propagation (Kendall et al., 2020) or variants (Dillavou et al., 2022; Anisetti et al., 2024) and they are universal function approximators (Scellier & Mishra, 2023). They leverage the principles of electrical circuits (e.g., Kirchhoff's and Ohm's laws) to perform inference and extract the weight gradients. Furthermore, the learning rules governing conductance changes are local. These features make nonlinear resistive networks good candidates as power-efficient learning-capable hardware, with recent experiments on memristive networks suggesting a potential 10,000x gain in energy efficiency compared to neural networks trained on GPUs (Yi et al., 2023). Small-scale networks based on these principles have been successfully built and trained on datasets like Iris (Dillavou et al., 2022), validating the approach.

Larger-scale simulations of nonlinear resistive networks on tasks such as MNIST (Kendall et al., 2020) and Fashion-MNIST (Watfa et al., 2023) further highlight their potential. However, these simulations, often conducted using the general-purpose SPICE circuit simulator (Keiter, 2014; Vogt et al., 2020), are hindered by SPICE's slowness. For example, Kendall et al. (2020) reported that simulating a one-hidden-layer network with 100 hidden nodes on MNIST took one week for just ten epochs of training. Due to the lack of efficient methods for simulating nonlinear resistive networks, some researchers have resorted to linear networks (Stern et al., 2022, 2024; Wycoff et al., 2022), thus missing an essential feature of machine learning: nonlinearity.

To address this, we introduce a novel methodology for simulating nonlinear resistive networks. Our algorithm, suitable for networks with arbitrary topologies, is an exact coordinate descent algorithm for convex quadratic programming (QP) problems with linear constraints (Wright, 2015). Applied to

*Second Workshop on Machine Learning with New Compute Paradigms at NeurIPS 2024 (MLNCP 2024).*

the 'deep resistive network' architecture of Kendall et al. (2020), our algorithm becomes an 'exact block coordinate descent' algorithm, whose run-time on GPUs is orders of magnitude faster than SPICE. The key contributions of this manuscript are:

- We show that, in a nonlinear resistive network, under an assumption of ideality of the circuit elements, the steady state configuration of node electrical potentials is the solution of a convex minimization problem: specifically, a convex quadratic programming (QP) problem with linear inequality constraints (Theorem 1).

- Using the QP formulation, we derive an algorithm to compute the steady state of an ideal nonlinear resistive network (Theorem 2). Our algorithm, which is an instance of an 'exact coordinate descent' algorithm, is applicable to networks with arbitrary topologies.

- For a specific class of nonlinear resistive networks called 'deep resistive networks' (DRNs) (Kendall et al., 2020), we derive a specialized algorithm to compute the steady state (Section 3). It exploits the bipartite structure of the DRN to perform exact block coordinate descent, where half of the coordinates are updated in parallel at each step of the minimization process. Each step of our algorithm involves solely tensor multiplications, divisions, and clipping, making it ideal to execute on parallel computers such as GPUs.

- We perform simulations of DRNs, trained with equilibrium propagation (EP) on the MNIST dataset (Section 4). Compared to the SPICE-based simulations of Kendall et al. (2020), our DRNs have up to 327 times more parameters (variable resistors) and the training time per epoch is 160 shorter, resulting in a 50000x larger network-size-to-epoch-duration ratio. We also train DRNs of two and three hidden layers.

## 2   Nonlinear Resistive Networks

We study electrical circuits composed of voltage sources, current sources, linear resistors, and diodes, collectively referred to as nonlinear resistive networks. These networks are modeled as graphs, with branches containing individual elements. We denote the set of branches $\mathcal{B} = \mathcal{B}_{\mathrm{VS}} \cup \mathcal{B}_{\mathrm{CS}} \cup \mathcal{B}_{\mathrm{R}} \cup \mathcal{B}_{\mathrm{D}}$, to represent voltage sources, current sources, resistors, and diodes, respectively. For each branch $(j, k)$, the voltage across a voltage source is $v_{jk}^{\mathrm{VS}}$, the current through a current source is $i_{jk}^{\mathrm{CS}}$, and the conductance of a resistor is $g_{jk}$.

We assume that these circuit elements are ideal, with behavior determined by the current-voltage ($i$-$v$) characteristics in Figure 1. The network's steady state is defined by branch voltages and currents satisfying branch equations and Kirchhoff's laws. Under the assumption of ideality of the circuit elements, the steady state of a nonlinear resistive network is described as follows.

**Theorem 1** (Convex QP formulation). *Consider a nonlinear resistive network with $N$ nodes, and denote $v = (v_1, v_2, \ldots, v_N)$ the vector of node electrical potentials. Under the assumption of ideality, the steady state configuration of node electrical potentials, denoted $v_\star$, satisfies*

$$v_\star = \arg\min_{v \in \mathcal{S}} E(v), \tag{1}$$

*where $E : \mathbb{R}^N \to \mathbb{R}$ and $\mathcal{S}$ are defined as:*

$$E(v_1, \ldots, v_N) := \frac{1}{2} \sum_{(j,k) \in \mathcal{B}_{\mathrm{R}}} g_{jk} \left(v_j - v_k\right)^2 + \sum_{(j,k) \in \mathcal{B}_{\mathrm{CS}}} i_{jk}^{\mathrm{CS}} \left(v_j - v_k\right), \tag{2}$$

$$\mathcal{S} := \{(v_1, v_2, \ldots, v_N) \in \mathbb{R}^N, v_j \leq v_k \quad \forall (j,k) \in \mathcal{B}_{\mathrm{D}}, v_j = v_k + v_{jk}^{\mathrm{VS}} \quad \forall (j,k) \in \mathcal{B}_{\mathrm{VS}}\}. \tag{3}$$

Some comments are in order. First, $\mathcal{S}$ is the set of feasible configurations of node electrical potentials, and $E$ is the power dissipation function (sometimes also referred to as 'energy function' although this terminology is not physically accurate). Importantly, not all feasible configurations satisfy the laws of electrical circuit theory; Theorem 1 states that among all feasible configurations, the one that is physically realized (the steady state) is the configuration that minimizes $E$.

Second, the feasible set $\mathcal{S}$ is defined by linear constraints. The inequality $v_j \geq v_k$ ensures non-negative voltage across diodes ($v_j > v_k$ if the diode is in the off-state, and $v_j = v_k$ if the diode is in the on-state), and the equality $v_j = v_k + v_{jk}^{\mathrm{VS}}$ ensures fixed voltage across voltage sources. Certain

conditions on network topology must be met to ensure $\mathcal{S}$ is non-empty. For instance, a loop of voltage sources with non-zero net voltage would violate Kirchhoff's Voltage Law (KVL), rendering $\mathcal{S}$ empty.

Third, the function $E$ represents half the power dissipated in resistors, plus the power dissipated in the current sources. Theorem 1 extends the 'principle of minimum dissipated power', stating that in a network of ideal voltage sources and linear resistors, the physically realized configuration minimizes power dissipation in the resistors.

Finally, $E(v)$ is a convex function of the node electrical potentials $v$, being a quadratic form with a positive definite Hessian. The feasible set $\mathcal{S}$ is also convex, defined by linear constraints. Thus, the steady state is the solution to a convex quadratic programming (QP) problem.

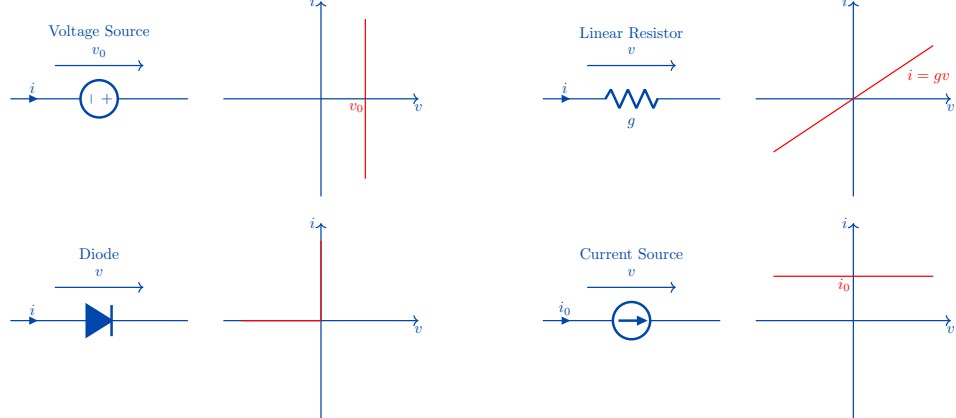

Figure 1: **Ideal circuit elements and their current-voltage (i-v) characteristics.** A linear resistor follows Ohm's law: $i = gv$, where $g$ is the conductance ($g = 1/r$, with $r$ being the resistance). An ideal diode operates in two states: "off" ($i = 0$ for $v \leq 0$) and "on" ($v = 0$ for $i > 0$). An ideal voltage source maintains a constant voltage $v_0$, independent of the current $i$, while an ideal current source maintains a constant current $i_0$, independent of the voltage $v$.

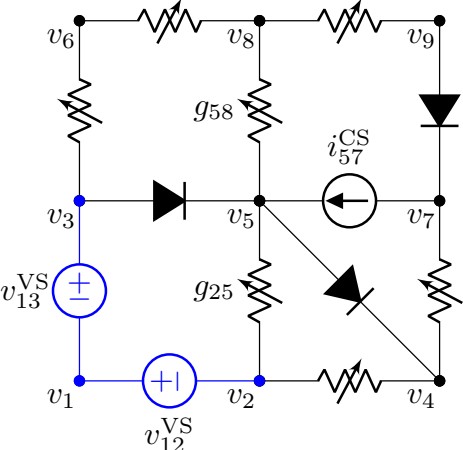

Figure 2: **A nonlinear resistive network.** The voltage sources form a tree (in blue), so if we set e.g. $v_1 = 0$, we can immediately infer $v_2 = -v_{12}^{\mathrm{VS}}$ and $v_3 = v_{13}^{\mathrm{VS}}$. The steady state is computed using exact coordinate descent (Theorem 2) on internal node electrical potentials (in black). For example, at node $k = 5$, $p_5$ is calculated from connected resistors and current sources, $p_5 = (g_{25}v_2 + g_{58}v_8 + i_{57}^{\mathrm{CS}})/(g_{25} + g_{58})$, and $v_5$ is determined by clipping $p_5$ within diode-imposed bounds, $v_5 = \max(v_3, \min(p_5, v_4))$. This process repeats for all nodes until convergence.

## 2.1 An Exact Coordinate Descent Algorithm to Simulate Nonlinear Resistive Networks

Using the QP formulation (Theorem 1), we derive a numerical method to compute the steady state $v_\star$ of an ideal nonlinear resistive network. Starting from an initial configuration $v^{(0)} \in \mathcal{S}$, our algorithm iteratively minimizes the function $E(v)$ by generating a sequence of configurations $v^{(1)}, v^{(2)}, \ldots$ within the feasible set $\mathcal{S}$, ensuring that $E(v^{(t+1)}) \leq E(v^{(t)})$ at each step.

Each step of the algorithm proceeds as follows. Let $v = (v_1, \ldots, v_N) \in \mathcal{S}$. First, we pick an internal node $k$ of the network. The function $E$ of Eq. (2) is a quadratic function of $v_k$ with other node electrical potentials fixed. Specifically, $E$ takes the form $E = a_k v_k^2 + b_k v_k + c_k$, where $a_k := \frac{1}{2} \sum_{j \in \mathcal{B}_R} g_{jk}$ and $b_k := -\sum_{j \in \mathcal{B}_R} g_{kj} v_j - \sum_{j \in \mathcal{B}_{CS}} i_{jk}^{CS}$. The function $E(v_k)$ is bounded below and its minimum in $\mathbb{R}$ is obtained at

$$p_k := -\frac{b_k}{2a_k} = \frac{\sum_{j \in \mathcal{B}_R} g_{kj} v_j + \sum_{j \in \mathcal{B}_{CS}} i_{jk}^{CS}}{\sum_{j \in \mathcal{B}_R} g_{kj}}. \tag{4}$$

The range of feasible values for $v_k$, however, is constrained by the diodes. It is of the form $[v_k^{\min}, v_k^{\max}]$. The minimum of $E(v_k)$ in this interval is found by clipping $p_k$ between $v_k^{\min}$ and $v_k^{\max}$. Specifically:

$$v_k^{\min} := \max_{j:(j,k) \in \mathcal{B}_D} v_j, \qquad v_k^{\max} := \min_{j:(k,j) \in \mathcal{B}_D} v_j, \qquad v_k' := \min\left(\max\left(v_k^{\min}, p_k\right), v_k^{\max}\right). \tag{5}$$

**Theorem 2** (Exact coordinate descent). *The configuration* $v' := (v_1, \ldots, v_{k-1}, v_k', v_{k+1}, \ldots, v_N)$ *minimizes the energy among all configurations* $v'' \in \mathcal{S}$ *of the form* $v'' = (v_1, \ldots, v_{k-1}, v_k'', v_{k+1}, \ldots, v_N)$, *i.e.,*

$$v_k' = \underset{v_k'' \,:\, (v_1, \ldots, v_{k-1}, v_k'', v_{k+1}, \ldots, v_N) \in \mathcal{S}}{\arg\min} E(v_1, \ldots, v_{k-1}, v_k'', v_{k+1}, \ldots, v_N). \tag{6}$$

*In particular* $v' \in \mathcal{S}$ *and* $E(v') \leq E(v)$.

Using Theorem 2, we minimize $E$ via an 'exact coordinate descent' strategy. Starting from a feasible configuration $v \in \mathcal{S}$, we iteratively select an internal node $k$, compute the value $v_k'$ that minimizes $E$, and update the configuration. This process is repeated until the energy $E$ converges to a minimum.

# 3 Deep Resistive Networks

The deep resistive network (DRN) model, introduced in Kendall et al. (2020), is a layered nonlinear resistive network inspired by neural networks. DRNs hold promise for hardware implementation with memristive crossbar arrays. In this section, we highlight the advantages of DRNs for simulations on multi-processor systems like GPUs.

## 3.1 Deep Resistive Network Architecture

In a DRN, a set of nodes connected to ground by voltage sources forms the input layer, while another set of nodes forms the output layer, whose electrical potentials represent model outputs.

**Power dissipation function.** DRNs are inspired by the architecture of multi-layer perceptrons (Figure 3). Each node in the network, also called a 'unit', has an electrical potential, analogous to a unit's activation in a neural network. We denote $v_k^{(\ell)}$ the electrical potential of the $k$-th node of layer $\ell$. Nodes in consecutive layers are interconnected by variable resistors, denoted $g_{jk}^{(\ell)}$, serving as trainable weights. The power dissipation function of a DRN is:

$$E(v) = \frac{1}{2} \sum_{\ell=1}^{L} \sum_{j=1}^{N_{\ell-1}} \sum_{k=1}^{N_\ell} g_{jk}^{(\ell)} \left(v_j^{(\ell-1)} - v_k^{(\ell)}\right)^2. \tag{7}$$

**Feasible set.** Nonlinearity in DRNs is introduced by placing diodes between unit nodes and ground, creating excitatory (non-negative potential) and inhibitory (non-positive potential) units. In hidden layers ($1 \leq \ell \leq L - 1$), even-indexed units are excitatory, and odd-indexed units are inhibitory. The output units ($\ell = L$) are linear, without diodes. The feasible set is:

$$\mathcal{S} = \{v \in \mathbb{R}^{\sum_{\ell=1}^{L} N_\ell} \mid v_k^{(\ell)} \geq 0 \text{ if k is even}, v_k^{(\ell)} \leq 0 \text{ if k is odd}, 1 \leq \ell \leq L - 1, 1 \leq k \leq N_\ell\}. \tag{8}$$

**Input voltage sources.** A key challenge in resistive networks is the non-negativity of weights: a conductance is non-negative. This is partially addressed by using both excitatory and inhibitory units in hidden layers. To further enhance representational capacity of the DRN, the number of input units is doubled to $N_0 = 2\dim(x)$, where $\dim(x)$ is the input dimension. The input voltage sources are set such that $v^{(0)}_{2k} = -v^{(0)}_{2k-1}$ for each $1 \le k \le N_0$. Furthermore, to prevent voltage decay as network depth increases, input voltages are amplified by a factor $A \gg 1$, setting $v^0_{2k-1} = +Ax_k$ and $v^0_{2k} = -Ax_k$.

**Output switches.** For equilibrium propagation (EP) learning, each output node is connected to ground via a switch, a resistor, and a voltage source used to set target values. These output resistors share a common conductance value $\beta > 0$. During inference, switches are open; during training, they close to drive output units toward target values.

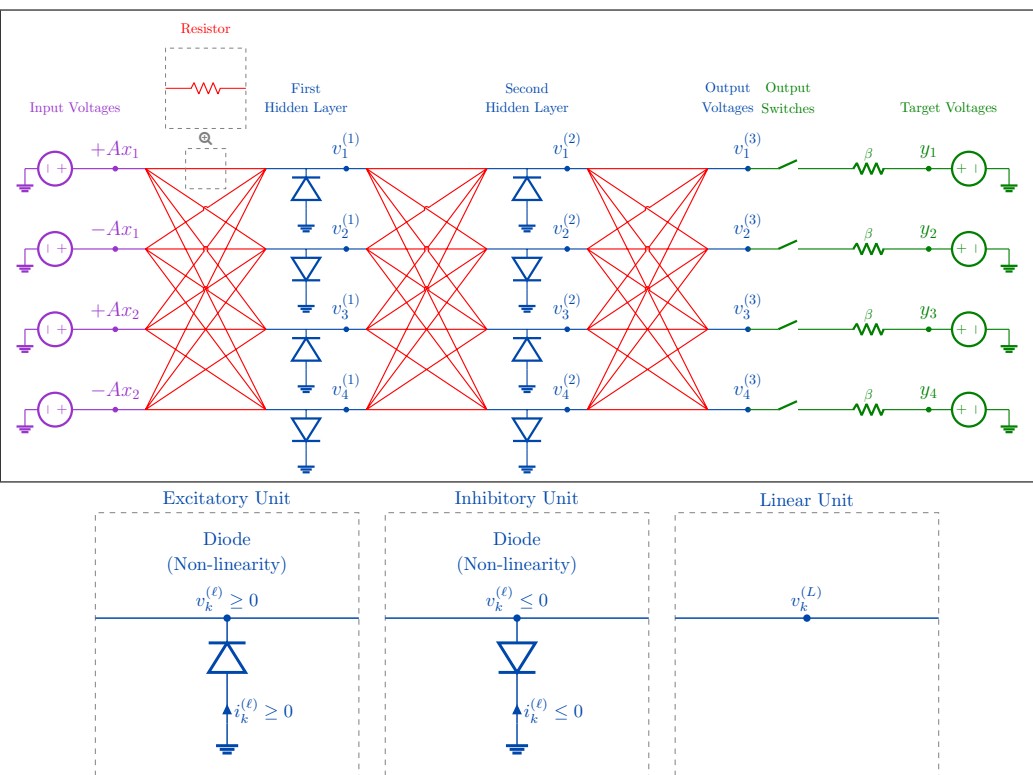

Figure 3: **Top.** A DRN with $L = 3$ layers. Input voltages are set according to $v^{(0)}_1 = Ax_1$, $v^{(0)}_2 = -Ax_1$, $v^{(0)}_3 = Ax_2$, and $v^{(0)}_4 = -Ax_2$, where $A$ is an amplification factor. During inference, output switches are open. During training (equilibrium propagation), switches close to nudge output nodes' electrical potentials $(v^{(3)}_1, v^{(3)}_2, v^{(3)}_3, v^{(3)}_4)$ towards target values $(y_1, y_2, y_3, y_4)$. In simulations, the update rule for a unit depends only on the states of units in adjacent layers (Eq. (9)), allowing simultaneous updates of even layers ($\ell = 2$) and then odd layers ($\ell = 1$ and $\ell = 3$). This process is called 'exact block coordinate descent'. **Bottom.** Diodes between nodes and ground create excitatory or inhibitory units depending on their orientation.

## 3.2 An Exact Block Coordinate Descent Algorithm to Simulate Deep Resistive Networks

In a DRN, the update rules (Theorem 2) for the 'hidden' units take the following form:

$$p_k^{(\ell)} := \frac{\sum_{j=1}^{N_{\ell-1}} g_{jk}^{(\ell)} v_j^{(\ell-1)} + \sum_{j=1}^{N_{\ell+1}} g_{kj}^{(\ell+1)} v_j^{(\ell+1)}}{\sum_{j=1}^{N_{\ell-1}} g_{jk}^{(\ell)} + \sum_{j=1}^{N_{\ell+1}} g_{kj}^{(\ell+1)}}, \qquad 1 \le \ell \le L-1, \qquad 1 \le k \le N_\ell, \quad (9)$$

$$v_k^{(\ell)} \leftarrow \begin{cases} \max\left(0, p_k^{(\ell)}\right) & \text{if k is even (excitatory unit)}, \\ \min\left(0, p_k^{(\ell)}\right) & \text{if k is odd (inhibitory unit)}. \end{cases} \qquad (10)$$

As for the output units, the update rule is:

$$v_k^{(L)} \leftarrow \frac{\sum_{j=1}^{N_{\ell-1}} g_{jk}^{(L)} v_j^{(L-1)} + \beta y_k}{\sum_{j=1}^{N_{\ell-1}} g_{jk}^{(L)} + \beta}, \qquad 1 \le k \le N_L \qquad (11)$$

where $\beta = 0$ if the output switches are open.

To speed up the computation of the DRN's steady state, we can exploit the fact that each $v_k^{(\ell)}$ update depends only on layers $\ell - 1$ and $\ell + 1$. Thus, all units in layer $\ell$ can be updated simultaneously. We obtain an 'exact block coordinate descent' algorithm. We can push this idea further, by partitioning the network into even-indexed and odd-indexed layers, and updating these groups alternately.

Equations (9) and (11) can be expressed in matrix-vector form, making each step of the algorithm involve $\sim L$ matrix-vector multiplications, $\sim L/2$ divisions, and $\sim L/2$ clipping operations. This makes the algorithm highly efficient for parallel computing platforms like GPUs.

## 4 Simulations

We used our exact block coordinate descent algorithm to train deep resistive networks (DRNs) with equilibrium propagation (EP) (Scellier & Bengio, 2017; Kendall et al., 2020) on the MNIST classification task. We trained DRNs with one, two, and three hidden layers (1024 units each), labeled DRN-1H, DRN-2H, and DRN-3H. Additionally, we trained two single-hidden-layer DRNs: the DRN-XS model (100 units) of Kendall et al. (2020), and the DRN-XL model (32,784 units), which is 327 times larger. Compared to the SPICE-based DRN-XS simulations of Kendall et al. (2020), the DRN-XL model was trained for 10 times as many epochs (100 vs 10), 16 times faster (10 hours and 27 minutes vs one week). Thus, our network-size-to-epoch-duration ratio is 50000x larger. The DRN-XL model also achieves a significantly better test error rate (1.33% vs 3.43%).

We also trained DRNs with backpropagation (BP) as a baseline. Although EP approaches the performance of BP, the small gap can likely be attributed to EP only approximating the gradient of the cost function, rather than extracting the exact gradient like BP does.

Table 1: Test error rates, epochs, and wall-clock time (WCT) for five DRN architectures (XS, XL, 1H, 2H, 3H) trained on MNIST using EP. Network sizes, as measured per the number of weights (in millions), are written in brackets. We report the SPICE-based results from Kendall et al. (2020) as a baseline (SPICE XS). We also trained DRNs using a version of backpropagation (BP) as a baseline for EP. The table includes mean and standard deviation values over five runs. Wall Clock Time (WCT) is expressed in HH:MM.

| DRN | EQUILIBRIUM PROPAGATION (EP) | | | BACKPROPAGATION (BP) | | |
| --- | --- | --- | --- | --- | --- | --- |
| | TEST (%) | EPOCHS | WCT | TEST (%) | EPOCHS | WCT |
| SPICE XS (0.16M) | 3.43 | 10 | 1 WEEK | | | |
| XS (0.16M) | $3.46 \pm 0.07$ | 10 | 0:30 | $3.30 \pm 0.13$ | 10 | 0:32 |
| XL (51.7M) | $1.33 \pm 0.02$ | 100 | 10:27 | $1.30 \pm 0.03$ | 100 | 8:19 |
| 1H (1.6M) | $1.57 \pm 0.07$ | 50 | 2:36 | $1.54 \pm 0.04$ | 50 | 2:48 |
| 2H (2.7M) | $1.48 \pm 0.05$ | 50 | 4:29 | $1.45 \pm 0.08$ | 50 | 4:53 |
| 3H (3.7M) | $1.66 \pm 0.09$ | 50 | 6:57 | $1.50 \pm 0.07$ | 50 | 7:55 |

Experimental details are available in the full version of the manuscript (Scellier, 2024), and the code for reproducing the results is available at `https://github.com/rain-neuromorphics/energy-based-learning`.

## 5 Discussion and Conclusion

Although this study focuses on layered architectures (DRNs), our algorithm applies to arbitrary network topologies, including unstructured (disordered) networks (Stern et al., 2022, 2024; Wycoff et al., 2022). Parallelization in such networks is also possible, though less straightforward to implement. Additionally, while we used equilibrium propagation (EP) for training, our exact coordinate descent algorithm is compatible with other learning methods, such as 'coupled learning' (Dillavou et al., 2022), 'frequency propagation' (Anisetti et al., 2024) and 'agnostic EP' (Scellier et al., 2022). More generally, by framing the steady state as a quadratic programming (QP) problem with linear constraints, our approach is compatible with various other QP-solving methods like primal-dual interior point methods (IPM) and sequential quadratic programming (SQP).

Our methodology to simulated nonlinear resistive networks presents limitations too. To simplify real-world complexities, it assumes ideal circuit elements (resistors, diodes, voltage sources, and current sources). Real diodes, however, exhibit non-ideal behaviors such as forward voltage drop and reverse leakage current. Despite these simplifications, our algorithm can be valuable for prototyping and exploring various network topologies. Future work could extend our approach to simulate networks with more realistic $i$-$v$ characteristics, such as piecewise linear curves. (However, it remains unclear whether smooth nonlinear characteristics such as exponentials could be efficiently and accurately simulated with this approach). Another caveat of the proposed DRN architecture is the need to amplify input voltages to prevent weight decay across the layers. This amplification can be very large, e.g. $A = 4000$ for the 3-hidden-layer DRN – see Scellier (2024) for the details of experiments. However, this can be mitigated by using 'bidirectional amplifiers' at each layer, instead of one large input gain (Kendall et al., 2020).

Looking ahead, our simulation methodology could significantly accelerate research in nonlinear resistive networks. It enables large-scale simulations of deep resistive networks (DRNs), providing a pathway to assess the scalability of these analog self-learning machines on more complex tasks.

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
