# OpenReview forum: "A fast algorithm to simulate nonlinear resistive networks"
_NeurIPS.cc/2024/Workshop/MLNCP — MLNCP Poster_

### Official Review · Reviewer_pGdG · 2024-09-19
**Paper suggests a new way of efficiently solving the responses of certain non-linear resistor circuits, providing for efficient computational implementation of learning algorithms in such systems.**

**Rating:** 9
**Confidence:** 5

**Review:**

Electronic circuits with adaptive resistor elements emerged recently in theory and experiment as a possible approach to implementing fast and energy efficient neuromorphic computing. However, computationally efficient ways of simulating such networks are lacking.
This paper suggests a quadratic programming approach (with linear inequality constraints) to efficiently compute the responses of these networks, containing certain ideal circuit elements including linear resistors and diodes. The addition of diodes as inequality constraints break the linearity of the network and allows it to compute, in principle, any nonlinear function. The authors explain how quadratic programming is useful in simulating such networks, implement coding tools for a certain topology reminiscent of deep neural networks, and demonstrate large speedups over previous computational approaches involving detailed circuit simulations (e.g. Spice).

This is overall an excellent paper that significantly extends the computational ability to simulate resistor-based learning circuits, and I can clearly see why this would be very useful for upcoming research on this topic. The ideas are clearly articulated and justified, showing why QP is natural for solving this problem. The simulated results also clearly indicate the computational advantage of this approach over previous ones. I also liked the authors idea of optimizing over the weights of odd and even indexed layers simultaneously. I am happy to recommend this paper for the MLNCP workshop.

I have some comments for the authors' consideration below:
1) Eq. 2 and following text: This is not an energy function but a power dissipation function, which is minimized due to Onsager’s law. While in theory this system behaves like an energy-based model, it actually isn’t really based on energy minimization. This is somewhat important as it means the network requires constant energy input to operate, as opposed to an actual thermodynamic “energy-based” system.
2) For the general coordinate descent algorithm, would you randomly pick an internal node potential at each iteration? Could one design a “smart” way of picking these nodes to accelerate convergence, i.e. by estimating which nodes require large changes to locally minimize power?
3) Is input amplification necessary in DRNs? If resistance values are small, I’d expect only mild voltage decay across layers.
4) Table and preceding text: Was it expected that backpropagation would give different results compared to equilibrium propagation? Both compute the same gradient and should scale similarly in terms of runtime. Also, please indicate units for wall clock time.

---

### Official Review · Reviewer_QH22 · 2024-10-02
**A useful tool for simulating analog ML networks**

**Rating:** 7
**Confidence:** 3

**Review:**

This paper describes a means for simulating nonlinear resistive networks for performing machine learning in a way that vastly outpaces SPICE simulations. The results are clearly shown and the algorithm is well-described. It is easy to see this being useful for determining the ideal working properties of a prospective analog machine learning circuit.


I have only a few minor comments and questions:

An indication of the required ranges of conductance and of voltage (max/min) would be helpful to assess the feasibility of some the architectures mentioned.

Piecewise linear functions are mentioned in the discussion – are smooth nonlinear functions such as exponentials (a more accurate model of a diode) only feasible with this algorithm if approximated piecewise?

Would an optimizer like Adam be useful for speeding this optimization process? Do the authors select initial conditions they know to be close to the minimum (e.g. the solution to the previous training step)?

---

### Decision · Program_Chairs · 2024-10-10

Accept (Poster)